# One-shot Federated Learning via Synthetic Distiller-Distillate Communication

**Junyuan Zhang**[1,2]    **Songhua Liu**[1]    **Xinchao Wang**[1]*
National University of Singapore[1]    Beihang University[2]
junyuanpk@gmail.com,    songhua.liu@u.nus.edu,    xinchao@nus.edu.sg

## Abstract

One-shot Federated learning (FL) is a powerful technology facilitating collaborative training of machine learning models in a single round of communication. While its superiority lies in communication efficiency and privacy preservation compared to iterative FL, one-shot FL often compromises model performance. Prior research has primarily focused on employing data-free knowledge distillation to optimize data generators and ensemble models for better aggregating local knowledge into the server model. Prior research has primarily focused on employing data-free knowledge distillation to optimize data generators and ensemble models for better aggregating local knowledge into the server model. However, these methods typically struggle with data heterogeneity, where inconsistent local data distributions can cause teachers to provide misleading knowledge. Additionally, they may encounter scalability issues with complex datasets due to inherent two-step information loss: first, during local training (from data to model), and second, when transferring knowledge to the server model (from model to inversed data). In this paper, we propose FedSD2C, a novel and practical one-shot FL framework designed to address these challenges. FedSD2C introduces a distiller to synthesize informative distillates directly from local data to reduce information loss and proposes sharing synthetic distillates instead of inconsistent local models to tackle data heterogeneity. Our empirical results demonstrate that FedSD2C consistently outperforms other one-shot FL methods with more complex and real datasets, achieving up to 2.6 $\times$ the performance of the best baseline. Code: `https://github.com/Carkham/FedSD2C`

## 1   Introduction

Federated learning (FL) has emerged as a cutting-edge technology that enables training a global model across multiple clients without sharing their raw data [1]. Original FL requires multiple communication rounds for exchanging information between clients and servers. While this paradigm yields a better global model by frequent communication, such high communication costs along with the risk of connection drop errors make it impractical and intolerable in real-world FL applications [2, 3, 4]. Moreover, frequent communication poses security risks such as man-in-the-middle attacks [5] and privacy concerns [6].

To address these issues, one-shot FL [7] has been proposed, requiring only a single communication round, significantly reducing communication costs and concurrently diminishing vulnerability to malicious interception. Due to one communication round property, One-shot FL can also be easily scaled up to large-scale client scenarios, especially for cross-device settings [3]. Despite its sufficient benefits, the limitation of a single communication round makes one-shot FL fall short in accuracy compared to conventional multiple-round FL.

---

*Corresponding Author.

38th Conference on Neural Information Processing Systems (NeurIPS 2024).

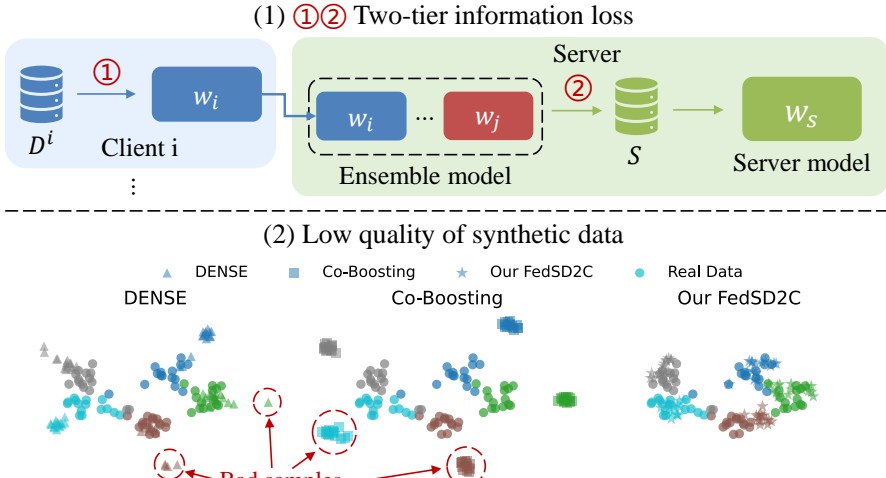

Figure 1: Illustration of issues in one-shot FL based on DFKD: (1) Information loss occurs during the transfer from local data to the model and from the model back to the inversed data. (2) t-SNE plots of feature distributions of data generated by DENSE(left ▲), Co-Boosting(middle ■), and our FedSD2C(right ★). We randomly select five different classes (indicated by different colors) of real and synthetic data from Tiny-ImageNet. Bad samples are data generated by the DFKD-based method that deviates from the distribution of local real data.

To facilitate effective knowledge transfer from client models to the server model within a single round, most previous one-shot FL methods [7, 8, 9, 10] have focused on knowledge distillation. Early approaches [11, 7] use knowledge distillation to transfer knowledge from an ensemble of client models [12] to the server model. While effective, these methods often require additional public datasets, which can be cumbersome or unfeasible in real-world scenarios [13]. Alternatively, data-free knowledge distillation (DFKD) is introduced to avoid the need for public datasets. For example, DENSE [8] employs Generative Adversarial Networks (GANs) [14] as data generators and an ensemble of client models as a discriminator to synthesize diverse data for knowledge transfer to server model in a data-free manner. Co-Boosting [10] extends DENSE by proposing a mutually reinforcing approach to enhance synthetic data and the ensemble model for server training.

Nevertheless, such a method of generating data implies two-tier information loss. First, due to model capacity limitations, client models may struggle to encapsulate all information about local data, affecting the quality of the generated data. Second, the generated data do not fully represent the information within the model, as they are produced from random noise without explicit guidance. Therefore, some classes of synthetic data cannot have similar feature distributions to the original real data, as depicted in Figure 1. Moreover, data heterogeneity [15] in FL can result in inconsistent and misleading predictions from local models [16], which has been shown to hinder knowledge distillation [17, 18]. Consequently, the server model trained on such noisy and information-lossy generated data typically suffers significant performance degradation, particularly on complex datasets [19].

In this paper, we propose FedSD2C (One-shot **Fed**erated Learning via **S**ynthetic **D**istiller-**D**istillate **C**ommunication), a novel and practical one-shot FL framework that introduces a pre-defined distiller for informative, privacy-enhanced, and communication-efficient distillate communication. In specific, FedSD2C first adopts a $\mathcal{V}$-information [20] based Core-Set selection method to distill the local dataset into an informative Core-Set. By capturing the diversity and realism through $\mathcal{V}$-information, the distilled Core-Set fully encapsulates the information of the local data domain for training a robust server model. However, directly transmitting the Core-Set, which may include the original samples, poses potential privacy risks and incurs significant communication costs, especially for high-resolution images. In this regard, FedSD2C employs two techniques to further distill the Core-Set into distillates, thereby enhancing privacy and reducing communication costs: 1) Utilizing Fourier transform perturbation to alter the amplitude components of the Core-Set samples for distillate initialization, enhancing privacy while retaining semantic content; 2) Employing a pre-trained Autoencoder [21] provided by the server as a distiller to distill the perturbed Core-Set into distillates and optimizing its $\mathcal{V}$-information to be as close as possible to original Core-Set, thus minimizing information loss. Finally, clients transmit synthetic distillates to the server instead of inconsistent

models for knowledge transfer. Compared to generating noisy knowledge from inconsistent client models with two-tier information loss, end-to-end distillate synthesis minimizes information loss and their aggregation mitigates the impact of data heterogeneity. Through extensive experiments over various real-world datasets , we show that our proposed method significantly surpasses the generated-based one-shot FL methods. The contributions of this paper are:

- We propose a new one-shot FL framework named FedSD2C which proposes to share synthetic distillates instead of generating noisy data from inconsistent models for server-side training.
- To mitigate the potential of privacy leakage and reduce communication costs, we propose two techniques: distillate initialization with Fourier transform perturbation and distillate synthesis with a pre-trained Autoencoder.
- We conduct extensive experiments over various datasets and settings. The results demonstrate the effectiveness of the proposed method which achieves up to $2.7 \times$ the performance of the best baseline.

## 2 Related Work

### 2.1 One-shot Federated Learning

One-shot federated learning was first proposed by [22], which introduces a method to aggregate a server model by distilling knowledge from an ensemble of client models using public datasets. FedKT [11] propose a hierarchical knowledge transfer framework, enabling various types of classification models. While their approaches demonstrate promising results, the requirement of public datasets which is inaccessible for privacy or transmission reasons limits their practical applications. To address the limitations associated with public datasets, DENSE [8] introduces a DFKD process utilizing an additional data generator trained on the ensemble model. Considering the challenge of high statistical heterogeneity, FedCVAE [9] proposes replacing the local training task with training conditional Variation Autoencoders. Furthermore, Co-Boosting [10] aims to enhance the performance of both the data generator and the ensemble model through a two-tier process. Despite these advancements, generating data through DFKD involves a two-tier information loss. Simultaneously, the inconsistency among client models due to data heterogeneity [15, 23, 24, 25], further degrades the quality of generated data, introducing label noise and thus limiting the performance of the server model. In this work, we tackle these problems from the perspective of sharing synthetic distillates. By utilizing Core-Set selection and pre-trained Autoencoders as distillers, our proposed methods distill diverse and informative data for server training.

### 2.2 Dataset Distillation in Federated learning

Dataset Distillation (DD) was first introduced by [26], aiming to distill the knowledge of datasets into synthetic data while preserving the performance of the model trained on it. Early dataset distillation methods are formulated as a bi-level optimization problems [27], where the outer loop optimize the synthetic data via gradient matching [28, 29, 30], distribution matching [31, 32] and performance matching [26], while the inner loop progressively trains a model on the synthetic data. Considering computational resources constraints, single-level optimization methods [33, 34] based on kernel ridge regression are proposed to decouple the bi-level optimization, thereby reducing training cost. These methods demonstrate comparable performance in non-complex datasets like CIFAR10 and are implemented in FL to tackle communication bottlenecks [35, 36], data heterogeneity [37, 38, 39] and one-shot FL [40, 36]. However, these methods require significant computational resources, making them impractical for edge devices with limited capability in FL. Additionally, they may struggle to effectively distill high-resolution datasets.

## 3 Methodology

### 3.1 Overview

We proposed FedSD2C to alleviate the two-tier information loss inherent in one-shot FL methods based on DFKD and account for data heterogeneity by synthetic distiller-distillate communication.

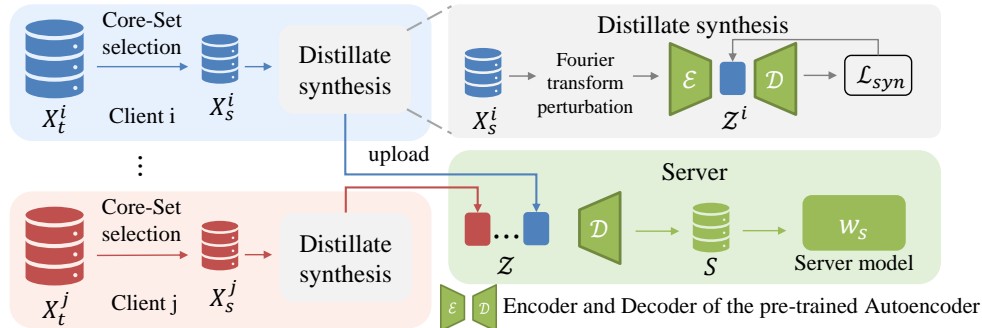

Figure 2: Framework of proposed FedSD2C.

The details of FedSD2C are described in Figure 2 and Algorithm 1. In the preparation phase of one-shot FL, the server distributed a pre-trained Autoencoder [21] as the distiller to each client. Subsequently, clients synthesize informative, privacy-enhanced, and communication-efficient distillates for server-side training. Specifically, to ensure the distillates fully encompass local information, clients first utilize a $\mathcal{V}$-information-based Core-Set selection method to extract diverse and informative Core-Set from their local data domains. Aiming to further reduce the communication costs and enhance the privacy of distillates, clients then perturb the Core-Set with Fourier transform for distillate initialization and employ the received pre-trained Autoencoder to optimize distillates in a compact latent space via $\mathcal{V}$-information alignment with the Core-Set. Finally, clients transmit the distillates, and the server decodes them using the pre-trained Autoencoder for training. We will now delve into the details of each component.

## 3.2 $\mathcal{V}$-information based Core-Set Selection

$\mathcal{V}$-information [20] was first proposed to measure the mutual information between $X$ and $Y$ constrained on predictive family $V$ which is denoted as:

$$I_{\mathcal{V}}(X \to Y) = H_{\mathcal{V}}(Y|\varnothing) - H_{\mathcal{V}}(Y|X) \tag{1}$$

where $H_{\mathcal{V}}(Y|\varnothing)$ and $H_{\mathcal{V}}(Y|X)$ denote the predictive $\mathcal{V}$-entropy conditioned on $\varnothing$ or $X$.

Core-Set selection is a type of dataset distillation method that focuses on preserving a subset of the original training dataset containing only valuable or representative samples. The objective is to enable models trained on this subset to achieve performance similar to those trained on the entire dataset. Typically, this is achieved by minimizing certain criteria, such as data distribution [41]. In our case, the emphasis lies on maximizing the diversity and information content of the subset. Therefore, in the context of $\mathcal{V}$-information, Core-Set selection can be reformulated as:

$$(X_s, Y_s) = \underset{X,Y}{\arg\max}\, I_{\mathcal{V}}(X_t \to Y_t) \tag{2}$$

where $X_t$ and $Y_t$ denotes the images and labels in the original datasets and $(X_s, Y_s)$ denotes the selected Core-Set. The intuition behind this equation is that Core-Set should include sufficient information and provide a concise representation corresponding to original datasets, constrained by observers $\mathcal{V}$.

Inspired by [42], we maximize the $\mathcal{V}$-information of the Core-Set from two levels. First, we identify the most informative image segments within each image by evaluating patches extracted at various scales from each image. Second, we select the top-$ipc$ with the highest $\mathcal{V}$-information for each class to construct the final Core-Set. The algorithm description can be found at Appendix A. Since a model pre-trained on original datasets can serve as an optimal observer for approximating the $\mathcal{V}$-information [42], we proposed using local pre-trained models as the observer models (predictive family $\mathcal{V}$) to conduct $\mathcal{V}$-information-based Core-Set selection on local datasets. Given that pre-trained local models are commonly present in one-shot FL [8, 7], their utilization does not violate the practicality of our approach.

## 3.3 Distilling Core-Set into Distillates with Pre-trained Autoencoders

In this section, we will describe how to synthesize informative, privacy-enhanced, and communication-efficient distillates using pre-trained Autoencoders. Although Core-Set significantly reduces dataset

size, communication costs remain a challenge for high-resolution data in real-world applications. Moreover, while sharing Core-Set provides consistent resistance to membership inference attacks [43], transmitting patches may still risk exposing sensitive information of original images. Therefore, further enhancing the privacy of shared data is essential. To alleviate these concerns, we propose two novel techniques: 1) distillate initialization with Fourier transform perturbation, which alters the amplitude components of the Core-Set samples to enhance privacy while retaining semantic content; and 2) distillate synthesis with pre-trained Autoencoders which act as the distillers. The pre-trained Autoencoder converts the perturbed samples into distillates by optimizing their $\mathcal{V}$-information to be as close as possible to the original Core-Set samples, minimizing information loss. We will now discuss each component in detail below. The process is described in Algorithm 1.

**Distillate initialization with Fourier transform perturbation.** A typical privacy-enhanced technique for sharing synthetic data is adding noise [44]. Although this approach can blur the visual information of the synthetic data, it can also destroy important semantic information and seriously degrade model performance. Our approach leverages a well-known property of the Fourier transform: the phase component of the Fourier spectrum encodes high-level semantic information, whereas the amplitude component captures low-level details [45, 46]. Inspired by this, we propose a novel privacy-enhanced method that perturbs the amplitude components in Core-Set samples through the Fourier transform to reduce visual information while preserving semantic information. Given an image $x$, its Fourier transform can be formulated as:

$$\mathcal{F}(x) = \mathcal{A}(x) \times e^{-j \times \mathcal{P}(x)} \tag{3}$$

where $\mathcal{A}(x), \mathcal{P}(x)$ depict the amplitude and phase components respectively. We then perturb the amplitude information via linearly interpolating:

$$\hat{\mathcal{A}}(x) = (1 - \lambda)\mathcal{A}(x) + \lambda \mathcal{A}(x^*) \tag{4}$$

where the $\lambda$ is a scaling coefficient and $x^*$ can be other images or random noise. Then, we combine the perturbed amplitude spectrums with the original phase component to generate the perturbed Core-Set sample:

$$x = \mathcal{F}^{-1}(\hat{\mathcal{A}}(x) \times e^{-j \times \mathcal{P}(x)}) \tag{5}$$

where $\mathcal{F}^{-1}(x)$ defines the inverse Fourier transform which can be calculated with the FFT algorithm [47] effectively.

**Distillate synthesis with pre-trained Autoencoders.** The Fourier transform perturbation serves to protect visual privacy but also compromises the realism of images, resulting in inconsistent $\mathcal{V}$-information between the perturbed Core-Set and the original Core-Set. To address this issue, we propose optimizing the alignment of the $\mathcal{V}$-information between the perturbed Core-Set and the original Core-Set to reconstruct key information. One straightforward and efficient approach is to optimize the perturbed Core-Set in pixel space [48, 49]. However, this method can be prone to overfitting into high-frequency patterns that only match the observer model [50]. Such overfitting is detrimental to training a global model due to inconsistent local models caused by data heterogeneity. Leveraging the powerful priors obtained from large-scale datasets, a pre-trained Autoencoder can decode latent representations into generalizable images. As a result, optimizing in the latent space acts as a regularization method that encourages synthetic data to be more generalizable, thereby making pre-trained Autoencoder an ideal distiller for local Core-Sets distillation. Furthermore, compact latent representations can reduce communication costs and mitigate privacy leakage if the latent is intercepted by attackers. Consequently, we employ a pre-trained Autoencoder on the client to distill the Core-Set into informative, privacy-enhanced, and communication-efficient distillates, and transmit them to the server for model training. On the server side, they are decoded by the decoder and are expected to maintain similar $\mathcal{V}$-information to the original Core-Set from the perspective of the observer model while remaining visually unidentifiable.

In specific, in the preparation phase of one-shot FL, the server first distributes a pre-trained Autoencoder to $n$ clients, denoted as $\mathcal{E}$ and $\mathcal{D}$ for encoder and decoder, respectively. Each client $i$ then conducts $\mathcal{V}$-information-based selection to construct a Core-Set $(X_s^i, Y_s^i), i = 1, 2 \cdots, n$ with diverse information regarding the original local datasets. Subsequently, client $i$ learns a latent set $\mathcal{Z}^i = \{z_j\}_{j=1}^{|\mathcal{Z}^i|}$ initialized by $\{\mathcal{E}(x_j^i)\}_{j=1}^{|X_s^i|}, x_j^i \in X_s^i$ which is perturbed with Fourier transform, such that the $\{\mathcal{D}(z_j^i)\}_{j=1}^{|\mathcal{Z}^i|}$ is as close as possible to the corresponding data in the Core-Set:

$$\underset{\mathcal{Z}^i}{\arg\min} \left\| I_{\mathcal{V}^i}(X_s^i \to Y_s^i) - I_{\mathcal{V}^i}(\{\mathcal{D}(z_j^i)\}_{j=1}^{|\mathcal{Z}^i|} \to Y_s^i) \right\|^2 \tag{6}$$

**Algorithm 1** One-shot Federated Learning via Synthetic Distiller-Distillate Communication

---

**Require:** Client local model $f^i(h^i(\cdot))$, pre-trained VAE $(\mathcal{D}, \mathcal{E})$, Server model parameter $\theta$, number of clients $n$, training iterations of local synthesis $T_{syn}$, training iterations of server model $T_{trn}$, learning rate $\eta_{syn}$ and $\eta_{trn}$.

1: Server distributes pre-trained VAE $(\mathcal{D}, \mathcal{E})$ to $n$ clients.
2: **for** each client $i = 1, \cdots, n$ **do**
3:      $(X_s^i, Y_s^i) \leftarrow$ CoresetSelection$(i)$            ▷ See Algorithm 2 in Appendix
4:      **for** each $x_j^i \in X_s^i$ **do**
5:          Perturb $x_j^i$ via Equations (3), (4) and (5)
6:      **end for**
7:      Initialize latent set $\mathcal{Z}^i = \{\mathcal{E}(x_j^i)\}_{j=1}^{|X_s^i|}$ using pre-trained VAE decoder $\mathcal{E}$
8:      **for** $t = 1, \cdots T_{syn}$ **do**
9:          **for** mini-batch $(z^i, x_s^i) \in (\mathcal{Z}^i, X_s^i)$ **do**
10:             Compute synthetic loss $\mathcal{L}_{syn}$ based on Equation (7)
11:             $z^i \leftarrow z^i - \eta_{syn}\nabla_{z^i}\mathcal{L}_{syn}$
12:          **end for**
13:      **end for**
14:      Generate soft label for each synthetic latents $Y_s^i = \{f^i(h^i(\mathcal{D}(z_j^i)))\}_{j=1}^{|\mathcal{Z}^i|}$
15:      Transmit $S^i = (\mathcal{Z}^i, Y_s^i)$ to the server
16: **end for**
17: Combine client synthetic data into $S = (\mathcal{Z}, Y_s) = (\mathcal{Z}^1 \cup \cdots \cup \mathcal{Z}^n, Y_s^1 \cup \cdots \cup Y_s^n)$
18: **for** $t = 1, \cdots, T_{trn}$ **do**
19:      **for** mini-batch $(z, y) \in (\mathcal{Z}, Y_s)$ **do**
20:          Compute $\mathcal{L}_{trn}$ based on Equation (8)
21:          $\theta \leftarrow \theta - \eta_{trn}\nabla_\theta\mathcal{L}_{trn}$
22:      **end for**
23: **end for**

---

As Core-Set selection employs the local pre-trained models as the observer models $\mathcal{V}^i$, the reformulated Equation (6) and objective function can be formulated as:

$$\arg\min_{z^i} \left\| h_i(\mathcal{D}(z^i)) - h_i(x^i) \right\|^2$$

$$\mathcal{L}_{syn} = \left\| \frac{1}{N}\sum_{j=1}^{N} h_i(\mathcal{D}(z_j^i)) - \frac{1}{N}\sum_{j=1}^{N} h_i(x_j^i) \right\|^2 \tag{7}$$

where $h_i(\cdot)$ denotes the feature extractor of pre-trained local model of client $i$, and $x_j^i$ and $z_j^i$ are paired. By minimizing $\mathcal{L}_{syn}$, we synthesize a set of latent variables $\mathcal{Z}^i = \{z_j\}_{j=1}^{|\mathcal{Z}^i|}$ that contains diverse information of local data domains.

Finally, clients transmit the latent set $\mathcal{Z}^i$ along with the corresponding soft label $Y_s^i$ predicted by local models to the server. The server combines the synthetic local distillates from each client $S = (\mathcal{Z}, Y_s)$. It then uses decoder $\mathcal{D}$ to reconstruct the images from data $(z, y) \in (\mathcal{Z}, Y_s)$ and distills the knowledge by minimizing the following objective function:

$$\mathcal{L}_{trn} = \sum_{(z,y)\in(\mathcal{Z},Y_s)} KL(f(h(\mathcal{D}(z))), y) \tag{8}$$

where $f(h(\cdot))$ denotes the server model. By minimizing the KL loss, we can transfer the local knowledge in the distillate to the server model.

**Discussion on privacy.** We first consider whether an attacker can train a performant model with the intercepted distilled data and labels during transmission [40, 9]. Because the attacker cannot know that the distilled data is encoded by VAE, nor can the attacker access the pre-trained VAE encoder, which can be easily achieved by being predefined offline or via encryption, it is hard for the attacker to reproduce an effective model. For model inversion and membership inference attacks, according to [51], there has been no prior research has performed these attacks solely using distilled data and

Table 1: Accuracy of different one-shot FL methods over three datasets with ConvNet and ResNet-18. We vary the $\alpha = \{0.1, 0.3, 0.5\}$ to simulate different levels of data heterogeneity for Tiny-ImageNet and Imagenette and use pre-defined splits for OpenImage. "Central" means that clients send all their local data to the server for centralized training, representing the upper bound of model performance.

| Model | Methods | ImageNette | | | Tiny-ImageNet | | | OpenImage |
| --- | --- | --- | --- | --- | --- | --- | --- | --- |
| | | $\alpha = 0.1$ | $\alpha = 0.3$ | $\alpha = 0.5$ | $\alpha = 0.1$ | $\alpha = 0.3$ | $\alpha = 0.5$ | - |
| ConvNet | Central | | 89.60 | | | 49.73 | | 33.61 |
| | FedAVG | 10.68±0.23 | 10.04±0.10 | 9.83±0.27 | - | - | - | 3.08±0.17 |
| | F-DAFL | 44.95±0.72 | 52.23±0.23 | 58.34±0.55 | 5.25±0.41 | 8.89±0.61 | 10.28±0.10 | 3.36±0.56 |
| | DENSE | 42.09±0.68 | 48.64±1.91 | 54.74±0.75 | 11.45±0.08 | 14.69±0.48 | 15.15±0.22 | 7.00±0.84 |
| | Co-Boosting | 39.36±0.70 | 56.15±1.33 | 58.60±1.02 | 6.66±0.35 | 9.81±0.26 | 10.75±0.11 | 13.59±0.98 |
| | FedSD2C | 50.68±0.20 | 57.89±0.96 | 58.17±0.51 | 20.73±0.12 | 23.53±0.18 | 24.10±0.30 | 23.00±0.24 |
| ResNet-18 | Central | | 90.00 | | | 61.98 | | 34.17 |
| | FedAVG | 9.86±0.13 | 10.06±0.20 | 10.76±0.35 | - | - | - | 1.68±0.16 |
| | F-DAFL | 37.86±0.38 | 39.52±0.46 | 46.06±0.16 | 7.91±0.22 | 12.30±0.36 | 13.31±0.56 | 12.75±0.14 |
| | DENSE | 38.37±0.36 | 47.85±2.17 | 49.78±2.11 | 8.88±0.23 | 13.05±0.36 | 17.24±0.43 | 14.85±0.62 |
| | Co-Boosting | 27.06±0.61 | 28.53±0.86 | 30.53±1.12 | 10.29±0.43 | 14.35±0.93 | 16.39±0.59 | 9.52±1.52 |
| | FedSD2C | 47.52±0.51 | 53.69±0.17 | 55.90±0.53 | 26.83±0.10 | 29.92±0.37 | 31.66±0.85 | 22.69±0.14 |

labels and previous works [52, 36, 35, 43] have also revealed the advantage of dataset distillation in this regard. Therefore, we employ the synthetic data as the reconstructed samples for the evaluation of model inversion attacks. Furthermore, we compare our proposed Fourier transform perturbation with other privacy-enhanced techniques, including adding random noise to synthetic samples and data augment [53]. Experimental results can be found in Section 4.3.1.

# 4 Experiments

## 4.1 Experimental Setup

**Datasets and partitions.** We conduct experiments on three real-world image datasets with different ranges of resolution including Tiny-ImageNet [54], ImageNette [55], and OpenImage [56]. Tiny-ImageNet contains 10000 images of $64 \times 64$ resolution across 200 classes. ImageNette is a widely used subset of 10 classes from ImageNet-1K [57] with 9469 color images, resized to $128 \times 128$. OpenImage is a large-scale real-world vision dataset with over 9 million images of $256 \times 256$ resolution. To simulate data heterogeneity in real-world applications of one-shot FL, we use Dirichlet distribution to generate non-IID data to generate non-IID local data, as in [58] for Tiny-ImageNet and ImageNette. Specifically, for client $i$, we sample $p_k^i \ Dir(\alpha)$ to allocate a $p_k^i$ proportion of class $k$ to client $i$. The parameter $\alpha$ controls the degree of data heterogeneity, with smaller $\alpha$ indicating severe data heterogeneity. The $\alpha$ is set to 0.1 by default unless otherwise stated. For OpenImage, we randomly choose $n$ real-world clients from FedScale [59] and use their corresponding test sets to form global sets. We set the default number of clients $n$ to 10, unless otherwise specified.

**Baseline methods and Configurations.** We compare our proposed FedSD2C with existing methods: FedAVG [1], DENSE [8] and Co-Boosting [10]. Following [8, 10], we also introduce DAFL [60] with one-shot FL settings, denoted as F-DAFL. We use two different model architectures: ConvNet [28] and ResNet-18 [61] for all methods. In FedSD2C, the image per class $ipc$ is set to 50 for Tiny-ImageNet and ImageNette, and 10 for OpenImage. We set the Fourier transform coefficient $\lambda = 0.8$ and use a public pre-trained Autoencoder from Stable Diffussion [62] by default for all tasks. For distillate synthesis, we set $T_{syn} = 50$, $\eta_{syn} = 0.1$ by default. $\eta_{trn}$ is set to 0.2 for Tiny-ImageNet and 0.02 for ImageNette and OpenImage. More experimental details can be found in the Appendix.

## 4.2 Evaluation Results

To evaluate the effectiveness of our method, we conduct experiments under various non-IID settings with $\alpha = \{0.1, 0.3, 0.5\}$ for Tiny-ImageNet and Imagenette and pre-defined splits [59] for OpenImage. As illustrated in Table 1, our proposed FedSD2C surpasses all other methods in most settings. In particular, under extreme data heterogeneity($\alpha = 0.1$), FedSD2C achieves up to $1.3\times$, $2.6\times$, and $1.8\times$ the accuracy of the best baseline on ImageNette, Tiny-ImageNet and OpenImage, respectively. This superior performance is attributed to FedSD2C's approach of sharing synthetic

Table 2: Accuracy, PSNR and SSIM of FedSD2C combining different privacy-enhanced techniques. *Laplace* and *Gaussian* indicate adding corresponding noise into synthetic distillates without Fourier transform initialization. FedMix denotes averaging two real samples from Core-Set to synthesize data. "-" indicates no privacy-enhanced technique is combined.

| Privacy-preserving techniques | ImageNette | | | | Tiny-ImageNet | | | |
|---|---|---|---|---|---|---|---|---|
| | ConvNet↑ | ResNet-18↑ | PSNR↓ | SSIM↓ | ConvNet↑ | ResNet-18↑ | PSNR↓ | SSIM↓ |
| - | 51.87 | 51.82 | - | - | 22.62 | 28.29 | - | - |
| Ours($\lambda = 0.1$) | 51.26 | 50.55 | 23.48 | 73.20 | 22.03 | 28.22 | 20.54 | 54.89 |
| Ours($\lambda = 0.5$) | 51.36 | 48.97 | 19.97 | 64.23 | 21.77 | 28.09 | 18.06 | 44.18 |
| Ours($\lambda = 0.8$) | 50.68 | 47.52 | 16.42 | 50.80 | 20.85 | 26.83 | 16.95 | 35.89 |
| $Laplace(s = 0.2, p = 0.1)$ | 48.61 | 45.25 | 24.02 | 81.66 | 21.50 | 27.48 | 22.25 | 73.09 |
| $Gaussian(s = 0.2, p = 0.1)$ | 48.31 | 46.70 | 24.82 | 85.89 | 21.48 | 27.51 | 23.38 | 78.90 |
| $Laplace(s = 0.2, p = 0.2)$ | 45.61 | 38.01 | 20.05 | 73.13 | 19.32 | 23.66 | 19.99 | 64.51 |
| $Gaussian(s = 0.2, p = 0.2)$ | 45.81 | 38.09 | 20.30 | 76.11 | 19.32 | 23.52 | 20.35 | 68.56 |
| FedMix | 41.86 | 37.76 | 16.88 | 58.93 | 13.86 | 16.26 | 16.43 | 56.91 |

distillates rather than inconsistent local models, thereby mitigating the impact of data heterogeneity. Moreover, FedSD2C demonstrates the independence from model structures. In contrast, other methods struggle to adapt to different model structures and complex datasets. For instance, at $\alpha = 0.5$, Co-Boosting with ResNet-18 achieves only half the accuracy of ConvNet on ImageNette, whereas FedSD2C maintains consistent performance. This discrepancy arises because differences in model capacity affect their ability to condense local knowledge and the two-tier information loss during data generation increases the difficulty of transferring local knowledge to the server model, resulting in poor robustness to complex datasets and varied networks. In contrast, the shared distillates in FedSD2C are synthesized through end-to-end local distillation, mitigating information loss during knowledge transfer.

## 4.3 Analysis of Our Method

### 4.3.1 Privacy Evaluation

For privacy evaluation, we consider an *honest-but-curious* server attempting to reconstruct client data from distillates. We compare our proposed FedSD2C with other privacy-enhanced techniques for sharing synthetic data, including adding random noise [44, 63] and FedMix [53]. In the random noise approach, we incorporate it into FedSD2C by removing the Fourier transform perturbation and instead directly using Core-Set samples for initialization. We then add random noise to the synthetic distillates before transmitting them to the server, following the methods in [44, 63]. Specifically, Given latent $z$, a perturbation coefficient $p$, randomly generated noise $e$ and its scale parameter $s$, the data to be shared is formulated as $z = (1 - p)z + e \times s$. FedMix [53] proposes using linear interpolation of real samples to preserve privacy. In this approach, we synthesize data by averaging each two real samples from the Core-Set. For our proposed Fourier transform perturbation, we vary the $\lambda = \{0.1, 0.5, 0.8\}$ and observe the variations in performance and privacy protection. To quantitatively evaluate the privacy protection of the synthetic data, we employ the Peak Signal-to-Noise Ratio (PSNR) and Structure Similarity Index Measure (SSIM). A higher PSNR or SSIM value indicates greater similarity between the synthetic samples and the original samples, which implies more severe privacy leakage. We calculate the average PSNR and SSIM values of all the synthetic samples.

As depicted in Table 2, although FedMix provides better privacy protection, as evidenced by lower PSNR and SSIM values, it comes at the expense of significant performance degradation. The application of random noise requires a delicate balance between performance and privacy protection. For example, with a perturbation coefficient $p = 0.2$, it offers similar privacy protection to that of FedMix, but the performance drops approximately 10% compared to $p = 0.1$. However, the $p = 0.1$ setting increases the risk of privacy leakage. In comparison, the synthetic distillates generated by our proposed FedSD2C achieve comparable PSNR values with them. This suggests that our proposed Fourier transform perturbation offers effective privacy protection for the real data sample. Furthermore, FedSD2C consistently outperforms other methods in terms of accuracy while maintaining a minimal performance degradation compared to no privacy protection techniques. This indicates that FedSD2C strikes a balance between privacy preservation and performance. We also

Table 3: Comparison of communication costs and accuracy at $\alpha = 0.1$ with ResNet-18. Results highlighted in **bold** represent outcomes with default $ipc$ settings. Acc. and Comm. denote accuracy and communication costs, respectively.

| Method | $ipc$ | ImageNette | | Tiny-ImageNet | | $ipc$ | OpenImage | |
|---|---|---|---|---|---|---|---|---|
| | | Acc. | Comm. | Acc. | Comm. | | Acc. | Comm. |
| DENSE | - | 38.37 | 44MB | 8.88 | 44MB | - | 14.85 | 44MB |
| Co-Boosting | - | 27.06 | 44MB | 10.29 | 44MB | - | 7.00 | 44MB |
| FedSD2C w/o AE | 1 | 19.90 | 0.48MB | 3.60 | 2.2MB | 1 | 12.89 | 2.6MB |
| FedSD2C | 20 | 43.10 | 0.23MB | 23.23 | 1.1MB | 5 | 20.73 | 1.6MB |
| | **50** | **50.68** | **0.5MB** | **26.83** | **2.1MB** | **10** | **22.69** | **2.0MB** |
| | 80 | 56.13 | 0.73MB | 27.71 | 2.9MB | 15 | 23.49 | 2.7MB |

perform membership inference attacks on FedSD2C and other methods, please refer to Appendix C.2 for more details.

### 4.3.2 Scalability of Communication Efficiency

By employing Core-Set selection and communication-efficient distillate communication, our FedSD2C condenses local data to mere MBs, while the model trained on these condensed data exhibits comparable performance, as shown in Table 3. Specifically, the communication costs of FedSD2C for sharing synthetic distillate is at most 4% of that of sharing model. Notably, we exclude the communication costs of sending and receiving pre-trained Autoencoder, as this can be pre-defined offline, allowing for the reuse of multiple one-shot FL tasks. Given the considerable capacity for communication costs, we further investigate the scalability of communication efficiency and performance. Our experimental results demonstrate that increased data transmission enhances the diversity of compressed data, leading to further improvements in performance. By increasing $ipc$ from 20 to 80, the accuracy boosts by 13.03% and 4.48% on ImageNette and Tiny-ImageNet respectively. Additionally, we compare FedSD2C without Autoencoder at equivalent communication costs, where the absence of synthesized images limits performance to at best half of the default setting. The communication-efficiency of FedSD2C highlights its practicality in real-world applications.

### 4.3.3 Impact of Pre-trained Autoencoder

Pre-trained Autoencoders are typically trained on natural data domains [64], while practical applications of federated learning often involve a broader range of domains, such as medical images. This raises the question of whether pre-trained Autoencoders remain effective when applied to a different domain and whether our proposed FedSD2C can adapt to these differences. To investigate this, we evaluate performance using a medical dataset COVID-FL [65].

As shown in Figure 3a, using the default synthesis iteration of $T_{syn} = 50$ yields suboptimal results. By increasing $T_{syn}$ to

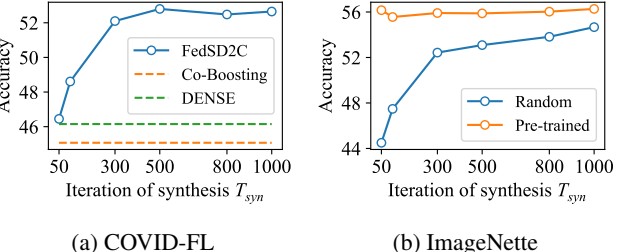

(a) COVID-FL      (b) ImageNette

Figure 3: (a) Experiments on the medical image data domain. Adopting pre-trained Autoencoders on other data domains can reduce performance. However, this can be mitigated by increasing $T_{syn}$. (b) Experiments of FedSD2C with randomly initialized downsampling and upsampling modules (blue line) compared to pre-trained Autoencoders (orange line) on ImageNette. Without pre-trained knowledge, FedSD2C requires a higher $T_{syn}$ for distillate synthesis but can still achieve comparable results. ResNet-18 is used for both experiments.

1000, the performance improves and then stabilizes. This observation suggests that the pre-trained knowledge of Autoencoders may influence the speed of distillate synthesis convergence. To further validate this, we replace the encoder and decoder of pre-trained Autoencoders with randomly initialized downsampling and upsampling modules. We vary $T_{syn}$ from 50 to 1000 and compare

Table 4: Accuracy on Tiny-ImageNet with different client amounts.

| Number of clients | $\alpha$ | F-DAFL | DENSE | Co-Boosting | FedSD2C |
|---|---|---|---|---|---|
| | $\alpha = 0.1$ | 4.97 | 11.36 | 7.37 | **21.92** |
| $n$=20 | $\alpha = 0.3$ | 7.66 | 13.69 | 9.95 | **22.00** |
| | $\alpha = 0.5$ | 10.01 | 14.86 | 10.45 | **22.87** |
| | $\alpha = 0.1$ | 3.99 | 8.32 | 7.05 | **21.48** |
| $n$=50 | $\alpha = 0.3$ | 5.92 | 12.25 | 8.80 | **22.38** |
| | $\alpha = 0.5$ | 6.94 | 13.06 | 8.90 | **22.58** |
| | $\alpha = 0.1$ | 3.12 | 7.87 | 4.27 | **20.34** |
| $n$=100 | $\alpha = 0.3$ | 5.26 | 10.22 | 7.15 | **21.86** |
| | $\alpha = 0.5$ | 6.30 | 11.49 | 8.32 | **21.76** |

its performance with employing pre-trained Autoencoders on ImageNette, setting $ipc = 80$ for better illustration. Figure 3b demonstrates that as $T_{syn}$ increases, the performance of FedSD2C with randomly initialized modules improves progressively, eventually matching the performance of FedSD2C with pre-trained Autoencoders. In summary, while pre-trained knowledge can enhance convergence rate, FedSD2C can achieve comparable performance by adjusting $T_{syn}$, demonstrating its adaptability across domains.

### 4.3.4 Impact of Client Scales

As practical FL deployments often involve participating clients [59], we evaluate our FedSD2C with various numbers of clients $n = \{20, 50, 100\}$ and maintain consistent communication budget by setting $ipc = \{40, 20, 10\}$, respectively. We compare these methods under on Tiny-ImageNet with data heterogeneity $\alpha = \{0.1, 0.3, 0.5\}$ for partitions and employ ConvNet. As depicted in Table 4, FedSD2C consistently achieves the highest accuracy as the number of clients increases. Moreover, FedSD2C demonstrates greater robustness to the number of participants. Specifically, as the number of participants changes, the accuracy of FedSD2C fluctuates within only 1%. In contrast, the accuracy of F-DAFL, DENSE, and Co-Boosting dropped by up to 4.71%, 3.49%, and 3.10%, respectively under different settings. This further validates the utility of sharing synthesized distillates in real-world one-shot FL applications.

## 5 Limitations

The local distillation process introduces additional computational overhead. While Core-Set selection requires no training and the distillate synthesis process only requires 50 iterations with a speed of 0.4s/per image on RTX3090, it still imposes higher resource requirements on the local device compared to the method of sharing model. One direction worth exploring is to integrate with the model market [66] to enable clients to synthesize distillates once for permanent use.

## 6 Conclusion

In this paper, we propose a new one-shot FL framework driven by distiller-distillate communication, denoted as FedSD2C, to alleviate the information loss of knowledge transfer and impacts of data heterogeneity. FedSD2C compresses the local data into Core-Set with $\mathcal{V}$-information and employs a pre-trained Autoencoder as the distiller to distill informative, communication-efficient, and privacy-enhanced distillates from Core-Set. Moreover, We discuss FedSD2C's resistance to attackers intercepting distillate communications and attacks from honest-but-curious servers and introduce Fourier transform perturbation to further minimize the risk of privacy leakage. Empirical results validate the effectiveness of FedSD2C in transferring local knowledge to the server in one-shot FL while balancing communication efficiency and privacy protection.

## Acknowledgment

This project is supported by the National Research Foundation, Singapore, under its AI Singapore Programme (AISG Award No: AISG2-RP-2021-023).

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

# A $\mathcal{V}$-information based Core-Set Selection Description

---

**Algorithm S1** $\mathcal{V}$-information-based Core-Set Selection

---

**Require:** Client local dataset $(X_t, Y_t)$ Client local feature extractor $h$, image per class $ipc$
    $X'_T \leftarrow \emptyset$
2: Level 1: identifying the most informative image segments
    **for** each $x, y \in (X_t, Y_t)$ **do**
4:    $\{x^k\}_{k=1}^K \leftarrow$ extract $K$ patches with multiple scales from $x$.
        **for** $k = 1$ **to** $K$ **do**
6:        Calculate $\mathcal{V}$-information score $s^k \leftarrow -\mathcal{L}(h(x^k), y)$
        **end for**
8:    select $x'$ with highest score $s'$
        $X'_T \bigcup \{(x', y, s')\}$
10: **end for**
    Level 2: select the top-$ipc$ patches with highest $\mathcal{V}$-information
12: $X_s, Y_s \leftarrow \emptyset$
    **for** each class $c \in Y_t$ **do**
14:    **if** size of $\{(x'_j, y_j, s'_j)\}, y_j = c \geq ipc$ **then**
        select top-$ipc$ $\{(x'_j, y_j)\}_{j=1}^{ipc}$ via $s'_j$
16:        $X_s \bigcup \{x'_j\}_{j=1}^{ipc}, Y_s \bigcup \{y_j\}_{j=1}^{ipc}$
    **end if**
18: **end for**
    **return** $(X_s, Y_s)$

---

# B  More Experimental Details

We use the SGD optimizer with momentum=0.9, learning rate=0.01 and weight decay=0.0001 for clients' local training. The batch size is set to 128 and local epoch is 200. For all generation based methods, we set the resolution of the generated images to $64 \times 64$, $128 \times 128$ and $256 \times 256$ for Tiny-ImageNet, ImageNette and OpenImage, the number of generated images in each batch is 128, and the learning rate of the generator is 0.001, the latent dimension is 256, iteration for training generator is 30, using Adam for optimization. The server model is optimized with SGD with momentum 0.9, the learning rate is 0.01, and the training epochs are 200. The synthesized batch size and server model training batch size is both 128. In DENSE, we set $\lambda_1 = 1$ for BN loss and $\lambda_2 = 0.5$ for diversity loss. In Co-Boosting, the perturbation strength is set to $\epsilon = 8/255$ and the step size $\mu = 0.1/n$. In Core-Set selection stage of FedSD2C, for each image $x_i$, we employ the `torchvision.transform.RandomResizeCrop` $K$ times to generate a collection of patches. For patch size, we set the scale=(0.08, 1.0), which is to collect diverse image patches. Following [42], we employ ConvNet-4 for Tiny-ImageNet, ConvNet-5 for ImageNette and ConvNet-6 for OpenImage. All methods are implemented with Pytorch and conducted on GeForce RTX 3090.

**Details of t-SNE plots in Figure 1** We randomly select a client and five classes from its local dataset (Tiny-ImageNet) and employs its local model (ResNet-18) to extract features. The feature is extracted from the final layer (before the classifier). We then use t-SNE plots to illustrate the feature distribution.

# C  Additional Experiments

## C.1  Additional datasets

**CIFAR-10.** We set the resolution of generated images to $32 \times 32$ for CIFAR-10 [67] and keep all the other settings the same. As shown in Table S1, there is an initial performance discrepancy at the standard setting of $ipc = 50$. This occurs because our method prioritizes efficiency with large datasets rather than low-resolution ones. However, upon increasing the amount of synthetic data ($ipc = 500$), our method can still achieve comparable results.

Table S1: Performance on CIFAR-10 with ResNet-18.

| Method | $\alpha = 0.1$ | $\alpha = 0.3$ | $\alpha = 0.5$ | Comm. |
|---|---|---|---|---|
| DENSE | 47.75 | 54.53 | 66.05 | 44MB |
| Co-Boosting | 53.33 | 61.75 | **68.99** | 44MB |
| FedSD2C ($ipc = 50$) | 44.72 | 48.23 | 51.14 | 0.03MB |
| FedSD2C ($ipc = 500$) | **56.95** | **62.37** | 65.06 | 0.24MB |

**COVID-FL.** We crop the images of COVID-FL [65] into 128×128, set the $T_{syn} = 1000$ as in Section 4.3.3 and keep all the other settings same. The results in Table S2 indicate that our FedSD2C still acheive better results compared to DENSE and Co-Boosting.

Table S2: Performance on COVID-FL datasets with ResNet-18.

| Method | $\alpha = 0.1$ | $\alpha = 0.3$ | $\alpha = 0.5$ |
|---|---|---|---|
| DENSE | 46.38 | 57.55 | 62.83 |
| Co-Boosting | 45.07 | 60.27 | 65.81 |
| FedSD2C | 52.65 | 62.50 | 66.68 |

## C.2  Membership Inference Attack

To further validate the effectiveness of our method, we employ an improved version of LiRA [68] to conduct Membership Inference Attacks on our methods. When attacking each client, for FedSD2C, we use the distillates uploaded by the client to train a new model and conduct membership inference attacks on that model. For the sharing model-based methods, we perform membership inference attacks on the models uploaded by the clients. The client model is ResNet-18 with $\alpha = 0.1, ipc = 50$. We set the raw images of Core-Set as the canary (target data $x$), as this is the most serious case of our methods. The results confirm that our approach does not introduce more privacy risk than the sharing model-based approach, even for the most vulnerable targets.

Table S3: Membership Inference Attack.

| Method | TPR@FPR=0.1% |
|---|---|
| Sharing model-based methods (DENSE, Co-Boosting) | 22.81 |
| FedSD2C | 20.13 |

## C.3  Wavelet transform perturbation

We explore the use of wavelet transforms to replace the Fourier transforms during Fourier transform perturbations. We use ResNet-18 on Tiny-ImageNet with $\alpha = 0.1, ipc = 50$. The results indicate that Wavelet Transform offers greater scalability in privacy protection. By increasing $\lambda$, the PSNR/SSIM can be reduced to as low as 12.90/15.30. When the accuracy is comparable to that of Fourier transform (Wavelet $\lambda = 0.5$ vs. Fourier $\lambda = 0.8$), the PSNR and SSIM of Wavelet transform is lower.

Table S4: Comparison between Wavelet transform and Fourier transform.

| | Acc. | PSNR | SSIM |
|---|---|---|---|
| Wavelet($\lambda = 0.1$) | 28.05 | 18.86 | 44.76 |
| Fourier($\lambda = 0.1$) | 28.22 | 20.54 | 51.50 |
| Wavelet($\lambda = 0.5$) | 26.91 | 15.22 | 27.34 |
| Fourier($\lambda = 0.5$) | 28.09 | 18.06 | 43.26 |
| Wavelet($\lambda = 0.8$) | 26.06 | 12.90 | 15.30 |
| Fourier($\lambda = 0.8$) | 26.83 | 16.95 | 35.89 |

## C.4  Ablation study on Core-Set selection

In this section, we perform ablation experiments to explore the significance of $\mathcal{V}$-information Core-Set selection. We use ResNet-18 with $\alpha = 0.1, ipc = 50$. Compared with $\mathcal{V}$-information Core-Set

selection, the accuracy of random selection decreased by 3.5 and 5.46 on Tiny-ImageNet and ImageNette, respectively. We also report the performance of uploading Core-Set directly, which achieve the best performance. However, without distillate synthesis, it will increase the cost of communication and the risk of privacy leakage.

Table S5: Performance of different selection strategy. Core-Set denotes that clients directly upload their local Core-Set, which leads to privacy issue. FedSD2C w/ random selection denotes replacing $\mathcal{V}$-information-based Core-Set selection with random selection.

|  | Tiny-ImageNet | ImageNette |
|---|---|---|
| Core-Set | 31.01 | 60.54 |
| FedSD2C w/ random selection | 23.32 | 42.06 |
| FedSD2C | 26.83 | 47.52 |

### C.5 Integrating with Differential Privacy

According to [35], introducing DP-SGD [69] during the distillate synthesis stage can provide theoretical privacy guarantees for our method. We perform experiments of integrating DP-SGD in our method on Tiny-ImageNet with ResNet-18 ($\alpha = 0.1, ipc = 50$) to provide a clear view of the trade-offs involved.

Table S6: Performance of integrating DP-SGD

|  | $\epsilon = 1$ | $\epsilon = 4$ | $\epsilon = 8$ | $\epsilon = \infty$ |
|---|---|---|---|---|
| FedSD2C | 22.92 | 25.13 | 26.01 | 26.83 |

## D Visualization

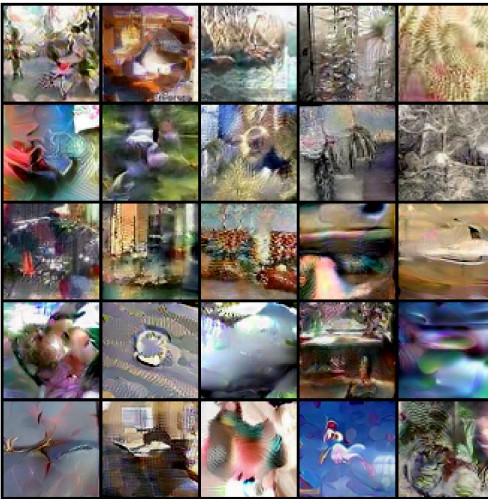 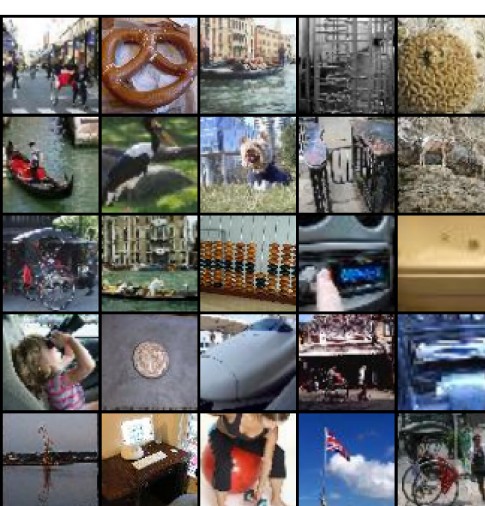

Figure S1: Visualization of synthetic distillate reconstructed by the pre-trained Autoencoder compared to the original sample on Tiny-ImageNet. The image style is similar, but with enhanced privacy protection.

