# OpenReview forum: "One-shot Federated Learning via Synthetic Distiller-Distillate Communication"
_NeurIPS.cc/2024/Conference — NeurIPS 2024 poster_

### Official Review · Reviewer_Bcom · 2024-07-07

**Soundness:** 3
**Presentation:** 2
**Contribution:** 2
**Rating:** 4
**Confidence:** 5

**Summary:**

The paper introduces FedSD2C, a new one-shot Federated Learning (FL) framework designed to address challenges in existing methods. Previous approaches have used data-free knowledge distillation to improve one-shot FL, but these methods struggle with data heterogeneity and scalability issues.  FedSD2C aims to solve these issues by:
1.Introducing a distiller to synthesize informative distillates directly from local data, reducing information loss.
2.Sharing synthetic distillates instead of inconsistent local models to address data heterogeneity.
The authors claim that empirical results show FedSD2C outperforms other one-shot FL methods, especially with more complex and real datasets.

**Strengths:**

This paper focuses on one-shot Federated Learning (FL), which is an intriguing topic.
The paper presents a comprehensive set of experiments, both on model performance and privacy.

**Weaknesses:**

Thanks for the authors' efforts in presenting this paper. After carefully reading it, I have the following questions and comments:

1. On line 52, the values 4.21 and 2.06 are not clearly explained. I suggest adding a figure or table to provide more specific details about these numbers.
2. While I can imagine that transmitting synthetic data could improve the global model's performance more easily than transmitting the model itself, this inevitably leads to privacy concerns. Although the authors provide some data reconstruction experiments, the results are not entirely convincing:
    a) To my knowledge, dataset distillation/coreset selection does not inherently protect privacy [1]. I suggest the authors include additional experiments on Membership Inference Attacks (MIA) (based on results from data-free KD [2], I suspect coresets provide even less privacy protection, since data-free KD doesn't use any 'real' training data ).
    b) According to the privacy onion concept [3], memorization is relative, then in your method the selected images may be at higher risk of privacy leakage compared to those not selected.

3. Compared to other methods, the proposed method communication overhead seems larger. It requires to transmit both the auto-encoder and a large amount of synthetic data.

4. The method relies heavily on Stable Diffusion's auto-encoder, which seems to utilize very strong prior information. How well would your method perform on datasets where diffusion models are not as proficient, such as medical datasets?

5. To my knowledge, another one-shot FL work[4] that relies on Stable Diffusion achieved results of about 75.0 on ImageNette, which appears significantly better than your 55.90.

[1] No Free Lunch in "Privacy for Free: How does Dataset Condensation Help Privacy"

[2] Evaluations of Machine Learning Privacy Defenses are Misleading

[3] The Privacy Onion Effect: Memorization is Relative

[4] Federated Generative Learning with Foundation Models

**Questions:**

see above

**Limitations:**

see above

---

> ### Author Rebuttal · Authors · 2024-08-06
>
> ***Q1 On line 52, the values 4.21 and 2.06 are not clearly explained. I suggest adding a figure or table to provide more specific details about these numbers.***
>
> We would like to thank the Reviewer Bcom for suggestions. We will include a detailed figure to better illustrate this in the final version.
>
> ***Q2 While I can imagine that transmitting synthetic data could improve the global model's performance more easily than transmitting the model itself, this inevitably leads to privacy concerns. Although the authors provide some data reconstruction experiments, the results are not entirely convincing: a) To my knowledge, dataset distillation/coreset selection does not inherently protect privacy [1]. I suggest the authors include additional experiments on Membership Inference Attacks (MIA) (based on results from data-free KD [2], I suspect coresets provide even less privacy protection, since data-free KD doesn't use any 'real' training data ). b) According to the privacy onion concept [3], memorization is relative, then in your method the selected images may be at higher risk of privacy leakage compared to those not selected.***
>
> Thanks for the insightful comments. As suggestion, we perform Membership Inference Attacks on our methods. We employ an improved version of LiRA[1]and set the raw images of Core-Set as the canary (target data $x$), as this is the most serious cases of our methods. Please note that we consider a semi-honest server, so the victim model for us is a model trained on synthetic data, and for the sharing model method is the client-uploaded local model. The results confirm that our approach does not introduce more privacy risk than the sharing model-based approach, even for the most vulnerable targets. Furthermore, according to Theorem 3.2 of [2], introducing DP-SGD during the distillate synthesis stage can provide theoretical privacy guarantees for our method.
>
> | Method | TPR@FPR=0.1 |
> | ---------------------------------- | ----------- |
> | Sharing Model (DENSE, Co-Boosting) | 22.81       |
> | FedSD2C                            | **20.13**      |
>
> [1] Aerni, M, et al. Evaluations of Machine Learning Privacy Defenses are Misleading. 2024.
>
> [2] Xiong Y, et al. FedDM: Iterative Distribution Matching for Communication-Efficient Federated Learning. 2023.
>
> ***Q3 Compared to other methods, the proposed method communication overhead seems larger. It requires to transmit both the auto-encoder and a large amount of synthetic data.***
>
> Thanks for the comments. In fact, our method **does not** require server-to-client transmission. The pre-trained Autoencoder is available from public repositories. Instead, in our framework, only the synthetic distillate is transmitted from the client to the server, with no other data or model being transferred. As illustrated in Table 3 of our paper, our method significantly reduces the communication cost to a mere 0.5 MB, in contrast to the 44 MB required by sharing model-based methods. This comparison underscores the superiority of our approach in communication efficiency.
>
> ***Q4 The method relies heavily on Stable Diffusion's auto-encoder, which seems to utilize very strong prior information. How well would your method perform on datasets where diffusion models are not as proficient, such as medical datasets?***
>
> Thanks for the detailed comments. The large-scale datasets used to pre-train the Autoencoder contain diverse data, which is enough to cover the data domains of most clients. For untrained data domains, such as medical dataset COVID-FL [1], we conducted experiments to verify that Autoencoder can be extended to more different data domains. All of these illustrate the practices of employing a pre-trained autoencoder. The experimental results on medical datasets are as follows:
>
> | Method | 0.1   | 0.3   | 0.5   |
> | ------- | ----- | ----- | ----- |
> | DENSE   | 46.15 | 57.55 | 62.83 |
> | CoBoost | 45.07 | 60.27 | 65.61 |
> | FedSD2C | **52.65** | **62.50** | **66.68** |
>
> [1] Yan R, et al. Label-Efficient Self-Supervised Federated Learning for Tackling Data Heterogeneity in Medical Imaging. 2023.
>
> ***Q5 To my knowledge, another one-shot FL work[4] that relies on Stable Diffusion achieved results of about 75.0 on ImageNette, which appears significantly better than your 55.90.***
>
> Thanks for bringing a related work to our attention. In fact, it is not absolutely fair to compare our approach to theirs. Firstly, the approach in work [1] demands significantly more computational resources and data. Their method requires 400 times the data volume compared to ours ($ipc=20,000$ vs. $ipc=50$). Our method is more cost-effective in terms of data synthesis. It relies solely on an autoencoder, eliminating the need for Stable Diffusion, which is a computationally intensive process. Secondly, the method from work [1] necessitates prior knowledge of the category name and assumes that the images adhere to the prior distribution of Stable Diffusion. This assumption can be restrictive, as it may not be applicable to diverse datasets or specialized domains such as medical imaging, where Stable Diffusion may not be able to synthesize X-rays or similar images based solely on labels.
>
> [1] Zhang J, et al. Federated Generative Learning with Foundation Models. 2023.

---

> > ### Comment · Reviewer_Bcom · 2024-08-12
> > **response**
> >
> > Thanks for authors' response.
> > As for the experiment on privacy attacks, since your model's performance is too low (far below 90%), it will also lead to biased MIA results. Therefore, I don't recommend claiming in the paper that your method can protect privacy (without any differential privacy guarantee).
> > For another one-shot FL work[1], I think they also tested the method on medical datasets.
> >
> > After reading the rebuttal, I am happy with increasing the score to a 5 or 6 (if there is no overstatement in the final version).
> >
> > [1] Zhang J, et al. Federated Generative Learning with Foundation Models. 2023.

---

> > > ### Author Response · Authors · 2024-08-14
> > > **A friendly reminder**
> > >
> > > Thanks for the feedback of Reviewer Bcom and we are encouraged that most of the concerns and questions are addressed. As mentioned by the reviewer, we will definitely include the results in the rebuttal into our revision.
> > >
> > > We have noted that the current rating still tends toward the negative. We kindly request clarification on any unresolved issues that might be affecting the reviewer's rating. Please feel free to share any remaining concerns. We are fully committed to addressing these issues during the remainder of the review discussion period. We greatly appreciate your efforts and look forward to your additional feedback.

---

> ### Author Response · Authors · 2024-08-12
> **Looking Forward to Further Discussions**
>
> Dear Reviewer Bcom,
>
> Thank you once again for your constructive comments and the effort you put into reviewing our submission. Please let us know if our response has addressed your concerns. We are more than happy to address any further comments you may have.
>
> Thanks!

---

> ### Author Response · Authors · 2024-08-12
> **Thank you for raising the score!**
>
> We sincerely appreciate the time and effort you have invested in reviewing our paper, and your insightful feedback has been a critical factor in enhancing the overall quality of our work.
>
> Following your comments, we will revise the privacy statement in the abstract and introduction to accurately reflect its limitations and ensure there is no overstatement. Additionally, we will include a discussion on work[1], as well as all other comments.
>
> Once more, we appreciate the time and effort you've dedicated to our paper.
>
> [1] Zhang J, et al. Federated Generative Learning with Foundation Models. 2023.

---

### Official Review · Reviewer_3gL8 · 2024-07-12

**Soundness:** 2
**Presentation:** 3
**Contribution:** 2
**Rating:** 4
**Confidence:** 3

**Summary:**

This paper proposes a one-shot FL method (FedSD2C), utilizing V-information to select local core set data and server-pretrained autoencoder and Fourier-domain perturbation to ensure privacy preservation for local "distillate" sharing. In comparison to existing works such as DENSE and Co-Boosting, FedSD2C can reduce information loss in one-shot FL and improve up to 2.7x global model performance.

**Strengths:**

The paper tackles an important setting in FL and the one-shot performance outperforms listed related works.

**Weaknesses:**

- The assumption that the server holds an autoencoder is pretty strong, as the autoencoder must be trained in the clients' data domain to ensure it works.

- The paper only showed that Fourier-based perturbation can 'visually' protect privacy by using PSNR as a metric. Although the paper mentioned MIA, it did not evaluate existing privacy attacks.

- The clarity of paper writing can be improved. Essential details are missing to understand the contributions. See more in the Questions section.

**Questions:**

- Can you more rigorously demonstrate the privacy-preservation statement? For example, differential privacy is a widely accepted concept. Could you please show the provable privacy-preservation guarantee like DP?

- It is unclear how Figure 1 is generated. The authors should detail the datasets, model, and necessary information. Otherwise, comparing the figures is meaningless.

- What is the computational cost for the core-set selection step?

- It is unclear which parts of the algorithm contribute to handling the heterogeneous data. It seems that coreset selection could contribute well if intentionally sampling (almost) balanced data.

- What is the number of clients for Table 1?

**Limitations:**

Limitation was discussed

---

> ### Author Rebuttal · Authors · 2024-08-06
>
> ***Q1 The assumption that the server holds an autoencoder is pretty strong, as the autoencoder must be trained in the clients' data domain to ensure it works.***
>
> Thanks for the detailed comments. The large-scale datasets used to pre-train the autoencoder contain diverse data, which is enough to cover the data domains of most clients. For untrained data domains, such as medical dataset COVID-FL [1], we conducted experiments to verify that Autoencoder can be extended to more different data domains. Moreover, Thanks to the growing open source community, autoencoders pre-trained for different data domains are also readily available. All of these illustrate the practices of employing a pre-trained autoencoder. The experimental results on medical datasets are as follows:
>
> | Method | 0.1   | 0.3   | 0.5   |
> | ------- | ----- | ----- | ----- |
> | DENSE   | 46.15 | 57.55 | 62.83 |
> | CoBoost | 45.07 | 60.27 | 65.61 |
> | FedSD2C | **52.65** | **62.50** | **66.68** |
>
> [1] Yan R, et al. Label-Efficient Self-Supervised Federated Learning for Tackling Data Heterogeneity in Medical Imaging. 2023.
>
> ***Q2 The paper only showed that Fourier-based perturbation can 'visually' protect privacy by using PSNR as a metric. Although the paper mentioned MIA, it did not evaluate existing privacy attacks. Can you more rigorously demonstrate the privacy-preservation statement? For example, differential privacy is a widely accepted concept. Could you please show the provable privacy-preservation guarantee like DP?***
>
> Thanks for the comments. Our paper emphasizes the empirical contributions of using the Fourier transform to enhance the privacy of synthetic data. In this regard, our paper performed Model Inversion Attacks to validate that our approach provides the best trade-off between privacy and performance.
>
> To further validate the effectiveness of our method, we employ an improved version of LiRA[1] to conduct Membership Inference Attacks on our methods. We set the raw images of Core-Set as the canary (target data $x$), as this is the most serious case of our methods. The results confirm that our approach does not introduce more privacy risk than the sharing model-based approach, even for the most vulnerable targets. Furthermore, according to Theorem 3.2 of [2], introducing DP-SGD during the distillate synthesis stage can provide theoretical privacy guarantees for our method.
>
> | Method | TPR@FPR=0.1 |
> | ---------------------------------- | ----------- |
> | Sharing Model (DENSE, Co-Boosting) | 22.81       |
> | FedSD2C                            | **20.13**      |
>
> [1] Aerni, M, et al. Evaluations of Machine Learning Privacy Defenses are Misleading. 2024.
>
> [2] Xiong Y, et al. FedDM: Iterative Distribution Matching for Communication-Efficient Federated Learning. 2023.
>
>
> ***Q3 Unclear statement of Figure 1 and What is the number of clients for Table 1?***
>
> We apologize for the unclear statement. We employ one of the local models (ResNet18) to extract features from its corresponding local dataset, and the synthetic data of DENSE, Co-Boosting and our proposed method. The feature is extracted from the final layer (before the classifier), and we use t-SNE plots to illustrate the feature distribution. This visual comparison aims to demonstrate the effectiveness of our method in capturing the data distribution. For all experiments presented in our paper, the default number of clients involved is 10, unless otherwise specified. We will include these details in our final version.
>
> ***Q4 What is the computational cost for the core-set selection step?***
>
> Thanks for the valuable question. In the Core-Set selection stage, the client runs $K$ (number of patch) inferences on each image $x_i\in X_t$ ($X_t$ indicates local dataset), i.e. the computational cost = $\mathrm{FLOPs}\times \|X_t\| \times K$
>
> ***Q5 It is unclear which parts of the algorithm contribute to handling the heterogeneous data. It seems that coreset selection could contribute well if intentionally sampling (almost) balanced data.***
>
> Thanks for the detailed comments. Data heterogeneity leads to inconsistencies in local models (known as client drift), which is a significant challenge for sharing model-based methods. Our method addresses this issue by transmitting synthetic distillate, which avoids the necessity for ensembling inconsistent local models and, consequently, mitigates the effect of client drift. Moreover, a pre-trained autoencoder is introduced to optimize distillate in its latent space. This manner prevents overfitting to patterns that are only recognized by the local model, which further mitigates the impact of data heterogeneity.

---

> ### Author Response · Authors · 2024-08-12
> **Looking Forward to Further Discussions**
>
> Dear Reviewer 3gL8,
>
> We would like to thank you once again for the insightful feedback and great efforts in reviewing our paper. Please let us know if you have follow-up concerns, and we are eager to engage in any further comments.
>
> Thanks!

---

> ### Comment · Reviewer_3gL8 · 2024-08-12
> **Thank you for the rebuttal**
>
> Thank you for the rebuttal, which partially addresses my original questions.
>
> It would be highly beneficial to delve deeper into the required capacity of an autoencoder, especially when it is trained on a domain different from the one being applied. The exploration of circumstances under which adaptation might fail, along with potential techniques to mitigate such failures or indicators that could signal these failures, is crucial. A deeper understanding of these underlying assumptions would significantly enhance the practical value of the proposed method.
>
> Additionally, I share the same concerns as Reviewer Bcom regarding the original submission's emphasis on privacy. Given that communication efficiency and privacy are highlighted as key contributions (contribution 2), revising the privacy-related statements could necessitate substantial changes to the original submission and might reduce overall contributions. I appreciate the inclusion of the new experiment with LiRA in the rebuttal. To further substantiate the privacy claims, it would be advisable for the authors to incorporate a more comprehensive empirical evaluation. Additionally, the rebuttal mentions that integrating differential DP into data generation could bolster privacy. However, it would be valuable to understand how this integration might impact model performance in practical scenarios.
>
> To clarify, my follow-up questions are not intended to request additional experiments but rather to better understand the merits and limitations of the work and to have a clearer picture of the revision plan.

---

> ### Author Response · Authors · 2024-08-14
> **Response**
>
> Dear Reviewer 3gL8,
>
> Thank you for your constructive feedback.
>
> 1. **Pre-trained Autoencoder**. We acknowledge that the effectiveness of a pre-trained autoencoder can be significantly reduced when applied to different data domains. When validating on the medical dataset COVID-FL, we set the number of iterations for distillate synthesis to $T_{syn}=1000$, which is much more than the $T_{syn}=50$ used for natural dataset, and observed a severe performance degradation when set to $T_{syn}=50$ for COVID-FL. We believe the primary impact may be on convergence speed. To illustrate, we conducted an experiment where we replaced the encoder and decoder of the pre-trained autoencoder with randomly initialized downsample and upsample layers, respectively, during distillate synthesis. The experiment is conducted on ImageNette with ResNet18 ($\alpha=0.1,ipc=80$). As illustrated in the table below, random initialization achieved comparable performance to the pre-trained Autoencoder after twenty times more iterations. The results illustrate two conclusion: 1) our method is not sensitive to the data domain and can achieve good results with more optimization iterations, and 2) the image prior in the pre-trained Autoencoder can speed up the optimization of distillate synthesis, and a slow convergence speed may be a potential indicator of the failure of a pre-trained autoencoder.
>
> | Method and Dataset                  | Accuracy | Iteration |
> | ----------------------------------- | -------- | --------- |
> | Pre-trained AE, COVID-FL          | 46.45    | 50        |
> | Pre-trained AE, COVID-FL          | 52.65    | 1000      |
> | Pre-trained AE, ImageNette        | 56.13    | 50        |
> | Random Initialization, ImageNette | 54.54    | 1000      |
>
> 2. **Privacy**. We acknowledge that our method does not provide provable privacy protection. Our primary emphasis is on empirical contributions, demonstrating that our method is superior in empirical evaluation compared to other baselines. As suggested by Reviewer nVDn, we have included the SSIM metric to further validate our method. Additionally, we perform experiments of integrating DP-SGD in our method on Tiny-ImageNet with ResNet18 ($\alpha=0.1,ipc=50$) to provide a clear view of the trade-offs involved. The results are shown below:
>
> |         | $\epsilon=1$ | $\epsilon=4$ | $\epsilon=8$ | $\epsilon=\infty$ |
> | ------- | ------------ | ------------ | ------------ | ----------------- |
> | FedSD2C | 22.92        | 25.13        | 26.01        | 26.83             |
>
> Thank you once again for your valuable comments. Your insights have been invaluable in refining our approach and understanding the practical implications of our work. We will include all these discussions in our final version.

---

### Official Review · Reviewer_EquD · 2024-07-12

**Soundness:** 2
**Presentation:** 2
**Contribution:** 2
**Rating:** 5
**Confidence:** 4

**Summary:**

The paper presents FedSD2C, a novel one-shot federated learning (FL) framework that aims to improve communication efficiency, privacy preservation, and model performance. The approach addresses issues with data heterogeneity and information loss by synthesizing informative distillates from local data and sharing these instead of local models. Empirical results show that FedSD2C significantly outperforms existing one-shot FL methods.

**Strengths:**

1. One-shot FL is a potential direction that can significantly minimize communication costs in FL.
2. This approach does not rely on sharing private data, which protects the data privacy.
3. Using V-information and fourier transform perturbation is interesting.
4. Experimental results show significant improvements.

**Weaknesses:**

1. A critical question on the optimal observer of approximating the V-information is that, the local model is trained locally, which may not indicate good V-information of the global datasets, i.e., all local dataets.
2. The privacy is not strictly guaranteed. The perturbation with 3, 4, 5 cannot ensure data privacy.
3. Figure 2 shows the reconstruction is very similar with the original images.
4. There is no theoretical analysis on privacy and generalization of the proposed method.
5. The pre-trained autoencoder plays a key role in the framework. But the experimental study doesn't investigate the impact of the pre-trained autoencoder on the performance.

**Questions:**

1. What are the key hyper-parameters used in the core-set selection algorithm?
2. According to Algorithm 2, the core-set includes patches with different scales. How does the encoder handle patches with different scales?
3. There is still a huge performance gap between FedSD2C and Central in Table 1. What are the main factors that result in the performance drop?

**Limitations:**

The authors have pointed out the computational overhead on local devices as the limitation.

---

> ### Author Rebuttal · Authors · 2024-08-06
>
> ***Q1 A critical question on the optimal observer of approximating the V-information is that, the local model is trained locally, which may not indicate good V-information of the global datasets, i.e., all local datasets.***
>
> Thanks for the detailed comments. In the context of one-shot federated learning, where global datasets are inaccessible, our method aims to distill optimal local datasets. It's essential to recognize that while local models are indeed tailored to their respective local data domains, this specialization is not a limitation but rather an advantage. This localized understanding positions the local model as the most effective observer for the Core-Set selection.
>
> ***Q2 The privacy is not strictly guaranteed. The perturbation with 3, 4, 5 cannot ensure data privacy. Figure 2 shows the reconstruction is very similar with the original images. There is no theoretical analysis on privacy and generalization of the proposed method.***
>
> Thanks for the detailed comments. Our paper emphasizes the empirical contributions of using the Fourier transform to enhance the privacy of synthetic data. In this regard, our paper performed Model Inversion Attacks to validate that our approach provides the best trade-off between privacy and performance.
> The image perturbations applied are designed to protect individual private information while still allowing for the synthesis of key image patterns. Therefore, similar image styles do not mean that private information has been compromised.
> To further validate the effectiveness of our method, we employ an improved version of LiRA[1] to conduct Membership Inference Attacks on our methods. We set the raw images of Core-Set as the canary (target data $x$), as this is the most serious case of our methods. The results confirm that our approach does not introduce more privacy risk than the sharing model-based approach, even for the most vulnerable targets. Furthermore, according to Theorem 3.2 of [2], introducing DP-SGD during the distillate synthesis stage can provide theoretical privacy guarantees for our method.
>
> | Method | TPR@FPR=0.1 |
> | ---------------------------------- | ----------- |
> | Sharing Model (DENSE, Co-Boosting) | 22.81       |
> | FedSD2C                            | **20.13**      |
>
> [1] Aerni, M, et al. Evaluations of Machine Learning Privacy Defenses are Misleading. 2024.
>
> [2] Xiong Y, et al. FedDM: Iterative Distribution Matching for Communication-Efficient Federated Learning. 2023.
>
> ***Q3 The pre-trained autoencoder plays a key role in the framework. But the experimental study doesn't investigate the impact of the pre-trained autoencoder on the performance.***
>
> Thanks for the detailed comments. In fact, we have explored the impact of pre-trained Autoencoders on communication efficiency in Table 3 of our paper. The results show that the introduction of pre-trained Autoencoders reduces the communication cost and achieves better results. To further validate its effectiveness, we perform experiments w/ and w/o Autoencoder during distillate synthesis ($ipc=50,\alpha=0.1$). The table clearly illustrates the performance gain, highlighting the advantage of pre-trained Autoencoders in our method.
>
> | Method | TinyImage | ImageNette |
> | -------------- | --------- | ---------- |
> | FedSD2C w/o AE | 24.35     | 46.43      |
> | FedSD2C        | 26.83     | 47.52      |
>
> In addition, we conducted experiments on the medical dataset COVID-FL [1] to verify that the employment of Autoencoder can be extended to more different data domains. The results are as follows:
>
> | Method | 0.1   | 0.3   | 0.5   |
> | ------- | ----- | ----- | ----- |
> | DENSE   | 46.15 | 57.55 | 62.83 |
> | CoBoost | 45.07 | 60.27 | 65.61 |
> | FedSD2C | **52.65** | **62.50** | **66.68** |
>
> [1] Yan R, et al. Label-Efficient Self-Supervised Federated Learning for Tackling Data Heterogeneity in Medical Imaging. 2023.
>
> ***Q4 What are the key hyper-parameters used in the core-set selection algorithm? According to Algorithm 2, the core-set includes patches with different scales. How does the encoder handle patches with different scales?***
>
> We apologize for the unclear statement. For each image $x_i$, we employ the `torchvision.transform.RandomResizeCrop` $K$ times to generate a collection of patches. Each patch with different scales will be resized to the resolution of its original image. We will revise this in the final version.
>
> ***Q5 There is still a huge performance gap between FedSD2C and Central in Table 1. What are the main factors that result in the performance drop?***
>
> Thanks for the question. This gap can be attributed to the inherent challenges of one-shot federated learning. For example, the data heterogeneity across different clients can lead to the generation of noisy soft labels, which can impede the server model from extracting accurate knowledge from the synthetic data.
> There is a natural trade-off between communication efficiency, which is crucial in federated learning to minimize communication overhead, and the performance of the server model, which requires sufficient information to learn effectively. Despite these challenges, it is important to recognize our method's strength. Our approach demonstrates the lowest performance gap and best communication efficiency among comparable methods, thereby offering optimal practical utility for one-shot federated learning environments. We believe that the trade-offs our approach entails are justified and favorable, given the current state of other methods and the inherent constraints of one-shot federated learning.

---

> > ### Comment · Reviewer_EquD · 2024-08-12
> > **Response to rebuttal**
> >
> > Thanks for the clarification and responses. I will rasie my rating to 5.

---

> ### Author Response · Authors · 2024-08-12
> **Looking Forward to Further Discussions**
>
> Dear Reviwer EquD,
>
> We would like to thank you again for your constructive comments and kind effort in reviewing our submission. We kindly ask you to inform us if our replies have successfully resolved your concerns, and we are more than happy to address any further comments.
>
> Thanks!

---

> ### Author Response · Authors · 2024-08-14
> **Thank you for raising the score!**
>
> Thank you very much for your acknowledgment of our rebuttal. We will include the results into our revision.

---

### Official Review · Reviewer_nVDn · 2024-07-12

**Soundness:** 2
**Presentation:** 2
**Contribution:** 2
**Rating:** 5
**Confidence:** 4

**Summary:**

This paper proposes a one-shot federated learning approach designed to enhance privacy protection, communication efficiency, and model performance. Firstly, the authors introduce a Core-Set selection method based on V-information to extract the most informative data from the original dataset. The amplitude spectrum of the images in the Core-Set is then perturbed using a Fourier transform, and these perturbed images are input into an Autoencoder to obtain their representations. Finally, these representations are transmitted to the server, which decodes the image information from the representations for training.

**Strengths:**

The one-shot learning method proposed in this paper enhances privacy protection, transmission efficiency, and model performance compared to previous methods. The approach appears innovative, employing theoretically grounded techniques such as the Core-Set selection method, and using an Autoencoder and decoder for data transmission, which improves both transmission efficiency and model performance.

**Weaknesses:**

1. In the Core-Set Selection stage, the authors do not explicitly define the patches used in Level 1: identifying the most informative image segments. I have the following questions: Are the patches the same size across different datasets? How many patches are extracted from each image? Does the size or number of patches affect the results? If so, how can one determine the optimal patch size?

2. In the perturbation stage, the authors do not provide sufficient reasoning for using the Fourier transform. Why must the Fourier transform be used instead of other transformations? For example, does using a wavelet transform and merging its low-frequency components achieve similar effects? Additionally, the authors mention that the merged image can be random noise. Is this too idealistic? Does using particularly weak random noise also achieve similar effects?

3. During the transmission phase using an Autoencoder, does the performance of the model depend on the pre-training of the Autoencoder? If a pre-trained model is not used, does this method still work effectively?

4. It would be beneficial for the authors to create a framework diagram of the entire method to enhance readability.


5. The paper contains many capitalization and punctuation errors, such as in line 50 and line 229. Additionally, please carefully review the citations in the article as they are quite disorganized, including issues with formatting, capitalization, and more. Please ensure consistency throughout.

**Questions:**

1. The paper utilizes several large datasets, but is the method effective for low-resolution datasets such as CIFAR-10?

2. The authors should provide the model performance using only the Core-Set, where clients send all their original Core-Set data to the server for centralized training.

3. Please provide additional visual metrics beyond PSNR to make the results more convincing, such as SSIM.

4. The authors should conduct ablation experiments to demonstrate the performance of each part of the proposed method. For example, evaluating the model performance without the Core-Set data selection step.

5. Since only representations are ultimately transmitted, and attackers cannot recover images from representations, I wonder if the image perturbation stage is still necessary?

**Limitations:**

Yes, have discussed the limitations

---

> ### Author Rebuttal · Authors · 2024-08-06
>
> ***Q1 Patch size, number of patch and influence of number of patch***
>
> **A1** We apologize for the unclear statement. For each image $x_i$, we employ the `torchvision.transform.RandomResizeCrop` $K$ times to generate a collection of patches. For patch size, we set the `scale=(0.08, 1.0)`, which is to collect diverse image patches, so we don't set a fixed size. For the number of patches, we perform experiments to determine the best $K$ empirically. The table below indicates that performance improves with an increased $K$, stabilizing when $K$ reaches 10. Consequently, we have empirically set $K=10$ based on these observations.
>
> | $K$          | 1     | 3    | 5     | 10    | 20    | 30    |
> | ------------ | ----- | ---- | ----- | ----- | ----- | ----- |
> | TinyImagenet | 16.58 | 19.8 | 21.66 | **23.34** | 23.38 | 23.29 |
> | ImageNette   | 52.89 | 54.8 | 55.06 | **55.13** | 54.06 | 53.45 |
>
> ***Q2 Motivation for Fourier transfomation. Merging random noise***
>
> We would like to thank Reviewer nVDn for the suggestion. The employment of the Fourier transform is for its ability to balance privacy preservation with information retention. Indead, wavelet transform is also suitable for our proposed methods. But our decision was driven by its simplicity. We appreciate the reviewer's insight and will consider exploring the wavelet transform and its potential benefits in future research.
>
> For image perturbations, we focus on preserving the frequency components of the Core-Set sample. The information in the amplitude component can be reconstructed in the latent space of the Autoencoder based on the image priors during synthesis. Since the noise does not follow the image a priori, merging with it will have a performance impact, but it can also lead to greater privacy protection. In response, we conducted an experiment to replace the merged images with Gaussian noise (ResNet18, $\alpha=0.1,ipc=50$, Tiny-ImageNet). The experimental results show that noise merging can be used as a supplement to the need for stronger privacy protection.
>
> | Method | Acc.  | PSNR  |
> | ---------------------- | ----- | ----- |
> | FedSD2C (Merging with Gaussian noise) | 22.21 | 12.91 |
> | FedSD2C                | 26.83 | 16.95 |
>
> **Q3 Effectiveness of Autoencoder, only Core-Set and performance of Core-Set selection step**
>
> 1) The introduction of pre-trained Autoencoders is crucial as it provides generalized image priors. This image prior helps synthetic distillate to prevent overfitting to localized data patterns, thus reducing the negative effects of data heterogeneity. The lower resolution of the latent space representations can significantly boost communication efficiency. Our experimental results presented in Table 3 of our paper, confirm the benefits of employing a pre-trained Autoencoder. Moreover, we conduct experiments **w/o Autoencoder** under the same setting (ResNet18, $\alpha=0.1,ipc=50$). The table clearly illustrates the effectiveness of our method even when not using a pre-trained model. However, the performance gain is substantial, highlighting the advantage of pre-training in our method.
> 2) Thanks for your feedback. We now add the results of only using Core-Set (ResNet18, $\alpha=0.1,ipc=50$) and perform an ablation study by replacing Core-Set selection with random selection. As depicted in the table, better performance can be achieved by simply transferring Core-Set, but this comes at the cost of privacy compromise. Random selection struggles to capture the necessary data diversity and representativeness, resulting in the poorest performance.
>
> | Method | Tiny-ImageNet | ImageNette |
> | --------------------------------------------- | --------- | ---------- |
> | Core-Set                                      | 31.01     | 60.54      |
> | FedSD2C w/o AE                                | 24.35     | 46.43      |
> | FedSD2C w/o Core-Set (Random selection) | 23.32     | 42.06      |
> | FedSD2C                                       | 26.83     | 47.52      |
>
> ***Q4 Experiments on CIFAR-10***
>
> Thanks for the pertinent comments. Our approach focuses on efficiency on large data rather than low-resolution datasets. Core-Set selection stage only requires inference, and distillate synthesis effectively reduces the number of parameters by optimizing in the latent space. This efficiency-tailored approach results in the synthesis of a more compact data compared to the DFKD-based methods. We have conducted a thorough evaluation of our approach on the CIFAR-10 dataset. As depicted in Table 2 (PDF), there is an initial performance discrepancy at the standard setting of $ipc=50$. However, upon increasing the amount of synthetic data ($ipc=500$), our method achieves results comparable results.
>
> ***Q5  Lack of SSIM.***
>
> As suggestions, we include the SSIM in our privacy evaluation. The results indicate that our method achieves the best tradeoff between performance and privacy protection.
>
> | Privacy-preserving | Acc.  | PSNR  | SSIM  |
> | ------------------------- | ----- | ----- | ----- |
> | ours ($\lambda=0.8$)      | 20.85 | 16.95 | 35.89 |
> | $Gaussian (s=0.2, p=0.2)$ | 19.32 | 23.52 | 68.56 |
> | $Gaussian (s=0.2, p=0.1)$ | 21.48 | 27.51 | 78.90 |
> | FedMix                    | 13.86 | 16.26 | 56.91 |
>
> ***Q6 The necessity of image perturbations***
>
> Although it is not possible for attackers to directly reconstruct images from their representations, there remains a risk of a semi-honest server attempting to infer private information from the data. By introducing image perturbation, we make it significantly more challenging for any adversary to deduce private information from synthetic distillate. Empirical results on the Model Inversion attack and Membership Inference attack (Table 2 of PDF file) demonstrate that our method achieves a superior balance between privacy protection and utility.
>
> ***Q7 Punctuation errors and framework diagram***
>
> We would like thank the Reviewer nVDn for suggestions. We will revise them in final version.

---

> > ### Comment · Reviewer_nVDn · 2024-08-12
> > **Thanks for the response**
> >
> > Dear authors,
> >
> > Thanks for your response. You have addressed most of my concerns, so I will raise the score to 5. However, one remaining concern is that you need to compare the Fourier transform and wavelet for different scenarios. For certain scenarios, wavelet may outperform the Fourier transform.

---

> > > ### Author Response · Authors · 2024-08-14
> > > **Response**
> > >
> > > Dear Reviewer nVDn,
> > >
> > > We would like to thank you for your constructive feedback.
> > >
> > > As suggestion, we perform experiments using the Wavelet Transform on Tiny-ImageNet with ResNet18($\alpha=0.1,ipc=50$). The results indicate that Wavelet Transform offers greater scalability in privacy protection. By increasing $\lambda$, the PSNR/SSIM can be reduced to as low as 12.90/15.30. When the accuracy is comparable to that of Fourier transform (Wavelet $\lambda=0.5$ vs. Fourier $\lambda=0.8$), the PSNR/SSIM of Wavelet transform is lower. We sincerely appreciate your innovative comments and will include this discussion in our final version.
> > >
> > > |                        | Acc.  | PSNR  | SSIM  |
> > > | ---------------------- | ----- | ----- | ----- |
> > > | Wavelet($\lambda=0.1$) | 28.05 | 18.86 | 44.76 |
> > > | Fourier($\lambda=0.1$) | 28.22 | 20.54 | 51.50 |
> > > | Wavelet($\lambda=0.5$) | 26.91 | 15.22 | 27.34 |
> > > | Fourier($\lambda=0.5$) | 28.09 | 18.06 | 43.26 |
> > > | Wavelet($\lambda=0.8$) | 26.06 | 12.90 | 15.30 |
> > > | Fourier($\lambda=0.8$) | 26.83 | 16.95 | 35.89 |

---

> ### Author Response · Authors · 2024-08-12
> **Looking Forward to Further Discussions**
>
> Dear Reviewer nVDn,
>
> We sincerely appreicate your great efforts in reviewing our submission. Your constructive comments really help improve our paper. Please do let us know if our response has addressed your concerns, and we are more than happy to address any further comments.
>
> Thanks!

---

### Author Rebuttal · Authors · 2024-08-06

Dear Reviewers and ACs,

We would like to thank the reviewers' insightful reviews and constructive comments on our manuscript. We have carefully considered all the suggestions and made the following changes:

1. We have included an additional datasets. By doing so, we aim to demonstrate our methods can adapt to low-resolution datasets (CIFAR-10) and vairous data domains (medical dataset COVID-FL[1])
2. To further substantiate the privacy-preserving capabilities of our proposed method, we have performed Membership Inference Attack and provided comprehensive explanations to mitigate privacy concerns.

Thank you once again for your valuable feedback.

[1] Yan R, et al. Label-Efficient Self-Supervised Federated Learning for Tackling Data Heterogeneity in Medical Imaging. 2023.

---

### Comment · Area_Chair_Xa4M · 2024-08-12
**Paper discussion**

Hi all,

Thanks again for your reviews!
If you haven't done so already, please respond to the rebuttals before the end of the author-reviewer discussion period. If you don't have any further questions for the authors then a simple "thank you for the rebuttal" would suffice.

All the best,
Area Chair

---

### Decision · Program_Chairs · 2024-09-25

**Decision:**

Accept (poster)

**Comment:**

This paper proposes a one-shot FL method (FedSD2C), utilizing V-information to select local core set data and server-pretrained autoencoder and Fourier-domain perturbation to ensure privacy preservation for local "distillate" sharing. In comparison to existing works such as DENSE and Co-Boosting, FedSD2C can reduce information loss in one-shot FL and improve up to 2.7x global model performance.

Weakness:

The biggest concern is the privacy protection. Although authors conduct experiments for MIA, but this is not sufficient to show the privacy protection.

•	The assumption that the server holds an autoencoder is pretty strong, as the autoencoder must be trained in the clients' data domain to ensure it works.

•	The paper only showed that Fourier-based perturbation can 'visually' protect privacy by using PSNR as a metric. Although the paper mentioned MIA, it did not evaluate existing privacy attacks. Given that communication efficiency and privacy are highlighted as key contributions (contribution 2), revising the privacy-related statements could necessitate substantial changes to the original submission and might reduce overall contributions.

•	As for the experiment on privacy attacks, since the model's performance is too low (far below 90%), it will also lead to biased MIA results. Therefore, it is not recommended to claim in the paper that your method can protect privacy (without any differential privacy guarantee).